# Sustainable energy management using the Internet of Things (IoT)

**Mian Hazrat Shah**[1,2]**, Shahbaz Khan**[2,3]***, Asif Khan**[2]**, Ilyas Khan**[4]***, Sayed M. Eldin**[5]

**1** Department of Industrial Engineering, University of Engineering and Technology, Peshawar, Pakistan, **2** Institute of Manufacturing, Engineering Management, University of Engineering and Applied Sciences, Islamabad, Pakistan, **3** Department of Mechatronics Engineering, University of Engineering and Technology, Peshawar, Pakistan, **4** Department of Mathematics, College of Science Al-Zulfi, Majmaah University, Al-Majmaah, Saudi Arabia, **5** Center of Research, Faculty of Engineering, Future University in Egypt, New Cairo, Egypt

* shahbazkhan@uetpeshawar.edu.pk (SK); i.said@mu.edu.sa (IK)

## Abstract

The world has a lot of want for energy due to the rapid pace of its consumption. The world's energy resources, especially non-renewable sources, are vanishing by leaps and bounds. However, agencies like the Paris Climate Agreement and the United Nations Sustainable development have defined some preventive measures to consider while consuming energy. The main issue in Pakistan is that the consumer is not supplied with electric power in a managed way, and the way of installation causes a lot of impairment to the expensive tools in the power distribution system. The motivation of this research focuses on energy management, making the distribution authority more powerful, digitalization, and protection of expensive components in electrical power systems. The proposed methodology uses current and voltage sensors to remotely monitor the amount of power being supplied to the consumer continuously, along with a microcontroller responsible for activating the relay in case of over-consumption and the Global System for Mobile (GSM) network to warn the consumer and inform the authority. This research work prevents manual and laborious meter readings and protects electrical instruments. Further, this work can enable online billing, pre-paid billing, and energy saving and provide a base for power theft detection.

## Introduction

Energy management is vital to a country's economic and industrial development [1]. Countries must fully utilize their available natural resources to reduce the consumption of resources and costs of electricity as much as possible [2]. Mostly, the electric energy of the entire world is generated in power plants by burning fossil fuels, which are non-renewable resources and produce adverse impacts on the environment [3]. Recent researches recommend that if fossil fuel consumption rates stay uniform and constant, the world reserve for fossil fuels will hardly last for 55–65 years [4]. Furthermore, the Paris Climate Agreement and the United Nations Sustainable Development have set goals to minimize environmental emissions [5–7]. If the power shortage crises remain imminent, modern civilization will be paralyzed. The tip of society will collapse.

**Data Availability Statement:** All relevant data are available within the paper and on Github: https://github.com/MIANHAZRATSHAH/Sustainable-Energy-Management-Using-IOT/blob/

48e3c37a980b47e405c240bdc89ae9ccb5bc7291/
Sustainable_Energy_Management_using_IOT.ino.

**Funding:** The funders had no role in study design, data collection and analysis, decision to publish, or preparation of the manuscript.

**Competing interests:** The authors have declared that no competing interests exist.

Also, in Pakistan, without proper energy and load management, excessive fuel sources can be easily observed [8]. Another issue in Pakistan is that the main service lines are connected directly to the consumers' meters without any protective measures, which causes severe losses and damages [9]. It threatens the most expensive elements in electric power distribution, such as transformers [10]. A power transformer, is shown in Fig 1 is considered the backbone and most critical equipment in electrical power conversion, so it requires better stability and reliable protection [11].

In Pakistan, only eight power-providing companies had damaged many power transformers. It results in a loss of almost 1.5 billion Pakistani Rupees. The Table 1 shows the number of transformers damaged in three months this year in those companies [12].

Unfortunately, every year in Pakistan shortage of electricity and no power availability is caused by the destruction of transformers as a result of overloading given in Fig 2, as well as creating a financial load on the treasure of the government, and no preventive measures are taken to prevent the loss [13]. It is impossible to achieve an immune electrical distribution system using the traditional manual operating and managing system [14]. Also, consumers in this modern world demand more cost-effective and reliable energy [15].

Furthermore, Pakistan has two types of energy meters: single-phase and three-phase energy meters [16]. In a single-phase energy meter, the consumer can use a maximum power of up to 5KW at a time. Similarly, in three phase energy meter, 15KW power is allowed to be used at a time [17]. A 100KVA transformer is set up for 20 single-phase energy meters [18]. There is no absolute check on the consumer's side to prevent excessive overloading, but when one of the consumers tends to exceed the threshold, the transformer is blown out [19] as shown in the Fig 3.

So, the proposed energy meter is designed for fixed power provision and can be easily graded according to the customer's demand. The consumer would be supposed to pay only by the power consumed. Everyone will be able to apply to the authority for its respective demands. The most significant feature of this energy meter is that no one can overload it. If any consumer dares to overload it, the energy meter will be tripped, generating two messages with the help of a GSM module: one to the consumer as a warning and the other to the regulatory authority for notification [20]. In such a way, the distribution transformer is protected, which is the core objective of this study. This research work aims to provide efficient and continuous readings from sensors to avoid billing errors and reduce maintenance costs. Empowering the authority to provide specific power, along with notification with the help of the GSM

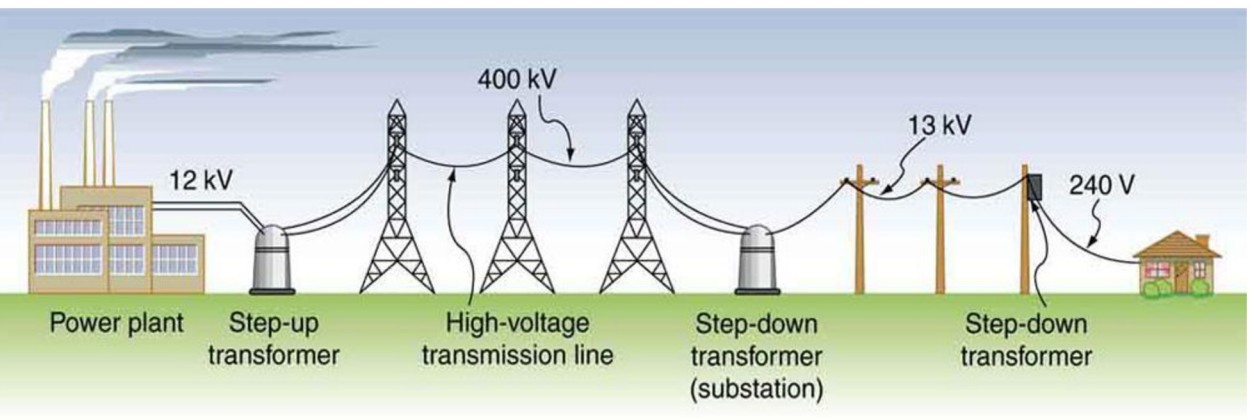

**Fig 1. Shows the role of transformer in power system.**

Table 1. The number of transformers damaged.

| S.No | Distribution Company | Number of Transformer Damaged |
|---|---|---|
| 01 | PESCO | 1110 |
| 02 | SEPCO | 434 |
| 03 | MEPCO | 915 |
| 04 | GESCO | 547 |
| 05 | FESCO | 610 |
| 06 | IESCO | 479 |
| 07 | KESCO | 202 |

network to consumers as they try to drag more power. The main contribution of this proposed research is to protect the expensive elements like transformer from consumer's side, while considering sustainability and not using chemicals, gases, or oil on transformer's side.

In short, this methodology will help:

- To make authority powerful to control the loads.

- To protect the transformer, a major and expensive component in the electrical system.

- To categorize the energy meter for different line rents.

- To eradicate labor consumption, as it is very tiresome in the traditional system.

- To prevent power theft.

- To put developing countries like Pakistan in the club of electrically-smart nations.

## Related work

With the invention of electricity, it was felt necessary to measure the amount of power consumed by an entity [21]. An instrument, an energy meter, was required to do the job. The

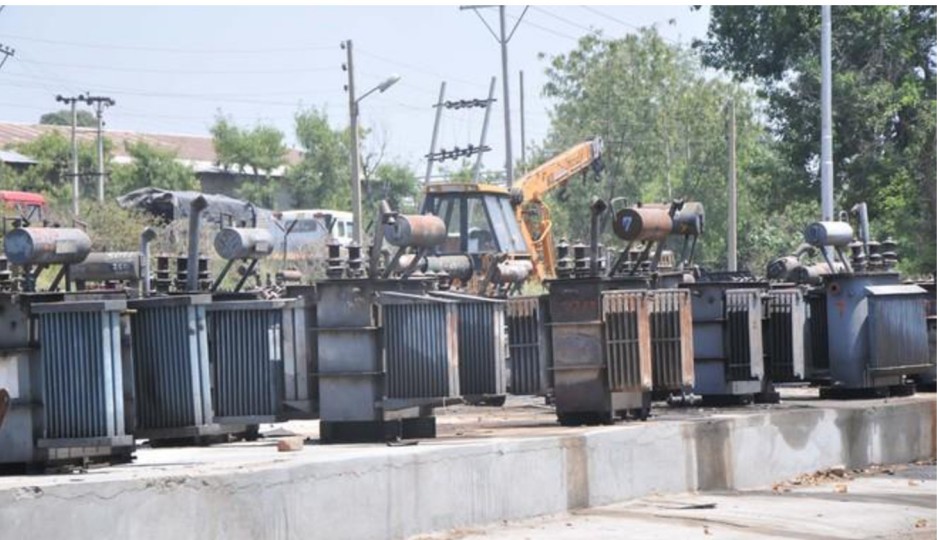

Fig 2. Damaged transformers.

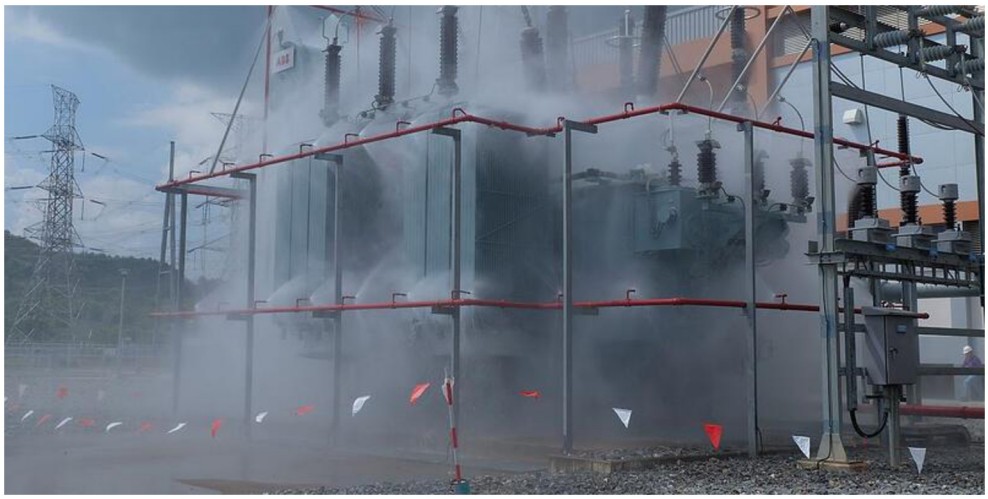

**Fig 3. A blown-out transformer.**

energy meter from then to now undergoes four generations: the first generation includes induction-type meters, also called mechanical meters. The second generation comprises electro-mechanical meters Fig 4, the third generation contains electronic meters, also known as static meters shown in Fig 5, and the fourth generation is the current smart meters [22].

Energy meters are made smarter for many purposes. Theodore Paraskevakos stood first in introducing The Automated Meter Reading (AMR) system while using advanced technology composed of wiring and wireless tools such as cable networks, power lines, RF modules, GSM modules and LCD for display purposes. M.V. Shinde and P.W. Kulkarni developed energy

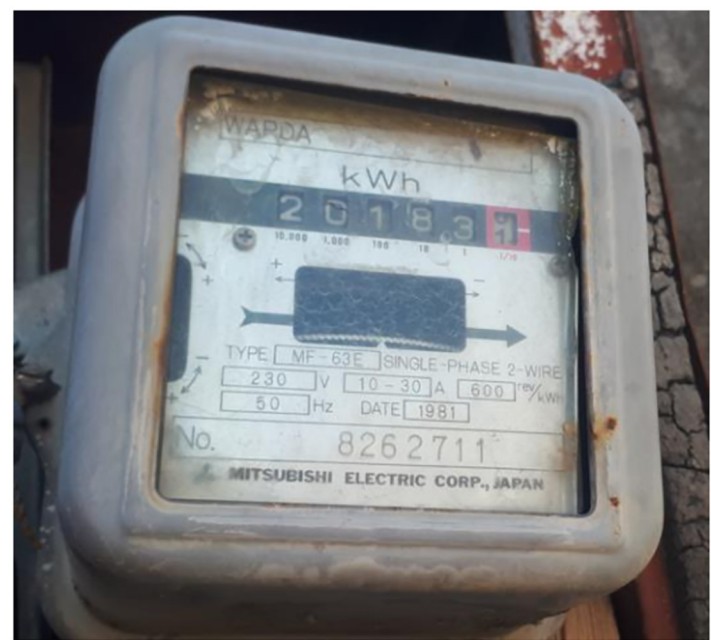

**Fig 4. Electro-mechanical meter.**

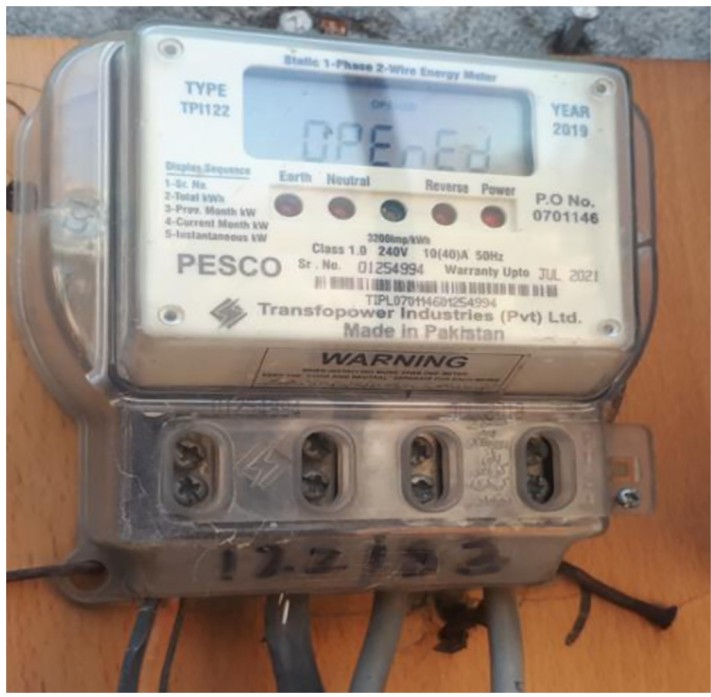

**Fig 5. Electronics meter.**

meter reading system using ZigBee and GSM architecture [23]. Y. B Mandak *et al.* developed an Automated Energy Meter that could monitor and control the reading remotely, but the energy measurement remained laborious and tiresome [24–26].

Md. Rahman *et al.* presented a smart energy meter for a superior and automatic measuring and billing system while integrating an Arduino module with a GSM modem. Their results show 74% efficiency [27]. Himshekhar Das and L.C. Saika proposed the same methodology with the integration of Virtual Instrumentation that can be operated in any computer for functioning as In-Home Display (IHD) for an Energy Management System (EMS). He supported his work by providing the data in Table 2 regarding the usage and feasibility of smart energy meters globally [28].

Anubhuti Anand *et al.* used the Internet of Things (IoT), sensors, LCD, GSM modules, simulation softwares and alarm systems to prevent and eradicate power theft and tempering issues [29–32]. M.A. Trivedi *et al.* considered the protection of the transformer necessary. They proposed a methodology consisting of a Knowledge-based Artificial Neural Network (ANN), Programmable Logic Controller (PLC), Cloud Data, GPS and GSM module [33–36].

A critical review of the literature has been summarized in the Table 3.

**Table 2. Implementation of smart meters across the globe.**

| S.No | Region | Number of Meter Deployed (In Million) | Number of Smart Meter Deployed (In Million) |
|------|--------|---------------------------------------|---------------------------------------------|
| 01 | USA | 150 | 46 |
| 02 | Europe | 280 | 61.2 |
| 03 | Canada | 15 | 7.3 |
| 04 | Asia | 435 | 260 |

**Table 3. Critical review of the literature.**

| S. No | Authors | Contribution | Components/Architectures | Limitations |
|---|---|---|---|---|
| 01 | R. Raju et al [33] | Transformer protection | Alarm, PLC, GSM and Temperature transducer | Implementation on transformer's side Based on temperature sensor not load |
| 02 | R.R Arabelli and D. Rajababu [34] | Transformer protection | Temperature and flow sensors, ESP32 and 2-Relay Module | Only applicable to oil filled and air insulated transformers Based on temperature sensor not load |
| 03 | M. V Shinde et al [23] | Development in energy metering | Zig-Bee, PLC, GSM, Camera, ARM LP2138 | Image transmission is costly Operating camera is laborious No record or database |
| 04 | Abhishek Pandey et al. [24] | Electric metering and billing | Optocoupler, Relay, Arduino, GSM | Only Pre-paid Metering No load restriction |
| 05 | J. Rahul et al. [36] | Billing and metering | Optocoupler, Real Time Clock (RTC), EEPROM, Relay, Arduino, GSM | Usage of extra components Costly |
| 06 | A. Anand et al. [29] | Power theft detection and load control | ESP32, Current sensor, ZMPT101B, Buzzer | Low Speed Inefficient |
| 07 | A. Fahim et al. [35] | Power theft detection, maintenance and monitoring | Arduino, weight sensor, Hall effect Sensor, IR sensor, Temperature sensor, Buzzer | Costly above $170 Manual disconnection |
| 08 | Z. Sultan et al. [25] | Remote monitoring, Controlling and Reduction of billing errors | PIC18F452, EEPROM, Real Time Clock (RTC), GSM | Auto disconnection and manual connection No load restriction Low efficiency |
| 09 | H. K. Patel et al. [26] | Billing error reduction and remote monitoring | LDR Sensor, RTC Timer, Arduino, Relay, GSM module | Low accuracy Buffering in data transmission |

So far, many objectives have been achieved in the proposed works and methodologies. But energy management is not catered to properly in terms of the usage of the Internet of Things (IOT), nor the consumer satisfaction ever considered, and the protection of the transformer from the consumer side is also not addressed. Therefore, this research aims to fill these gaps using Smart Energy Meter (SEM).

## Materials and methods

The proposed methodology as shown in the Fig 6 comprises an energy meter, current and voltage sensors, an Arduino microcontroller, a GSM module, a Relay module, and an LCD for

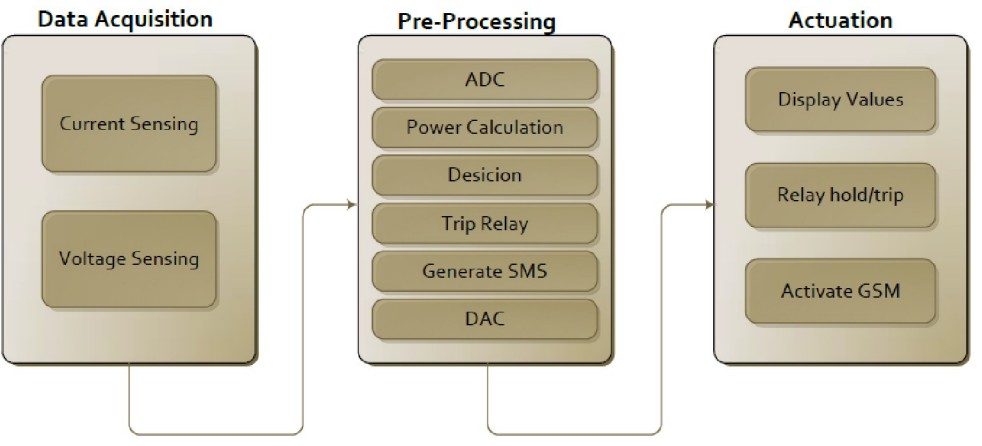

**Fig 6. Research methodology diagram.**

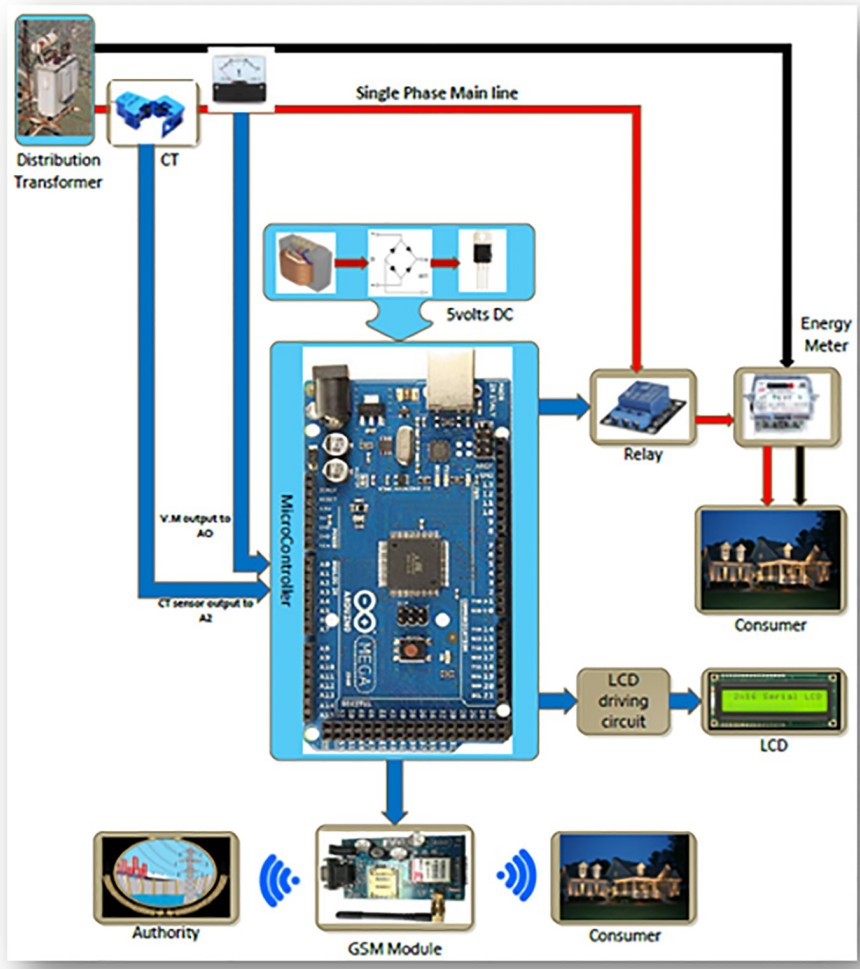

**Fig 7. Block diagram of proposed methodology.**

display. When the lines of the energy meter are energized, the current and voltage sensors read their value continuously to calculate the amount of power flow, as power is the product of both current flow and voltage applied. The Arduino microcontroller reads the values from the sensors and displays these values on LCD. The Arduino Microcontroller is the main element responsible for comparing the amount of power flow with the preset conditions. The GSM and Relay modules are connected to the Arduino microcontroller for actuation.

Whenever a customer tries to drive more power than the authority's allotted amount, the current and voltage sensors will input the data to the Arduino microcontroller. The microcontroller will compare these. If the dragged power is more, the microcontroller will trip the energy meter by actuating the Relay module and generating two text messages with the GSM module's help, as shown in Fig 7. One message, "Dear Customer, you have overloaded the transformer. Therefore, your meter is tripped off" will be sent to the customer. The other message, "The meter no. . .. is tripped off due to overloading the transformer" will be sent to the authority. As all customers do not need the same amount of power, they are all provided only

with 5KW and 15KW power in Pakistan. Here, the factor of energy management is introduced so that any customer can be provided with the desired amount of power, which will be considered the preset condition, and the customer is to be charged accordingly. So, the proposed energy meter can be graded based on customers' demands.

## Software simulation

The simulation is carried out with the help of Proteus Simulating Software using version 8 professional, as shown in Fig 8. Simulation in Proteus is very efficient and gives almost 100% realistic imitation of the work done physically. Similarly, the schematic diagram of the proposed methodology is done in Fritzing Software using Fritzing.0.9.3b.64.pc as provided in Fig 9.

In Fig 10, the flowchart of the proposed methodology has been given to highlight the complete working and logical process of the developed programming code of the proposed prototype. As shown in the flowchart, when the prototype is started, some steps are taken to initialize and obtain the data from the sensors that is injected into the microcontroller. After getting the data from sensors, the microcontroller continuously compares it with the preset limit and displays it on LCD. When the data contrasts with the preset limit, the microcontroller activates GSM and relay for their functions.

## Hardware implementation

The hardware circuit has been designed on a Printed Circuit board (PCB), as shown in the Fig 11. The following tools and component usage are described in detail.

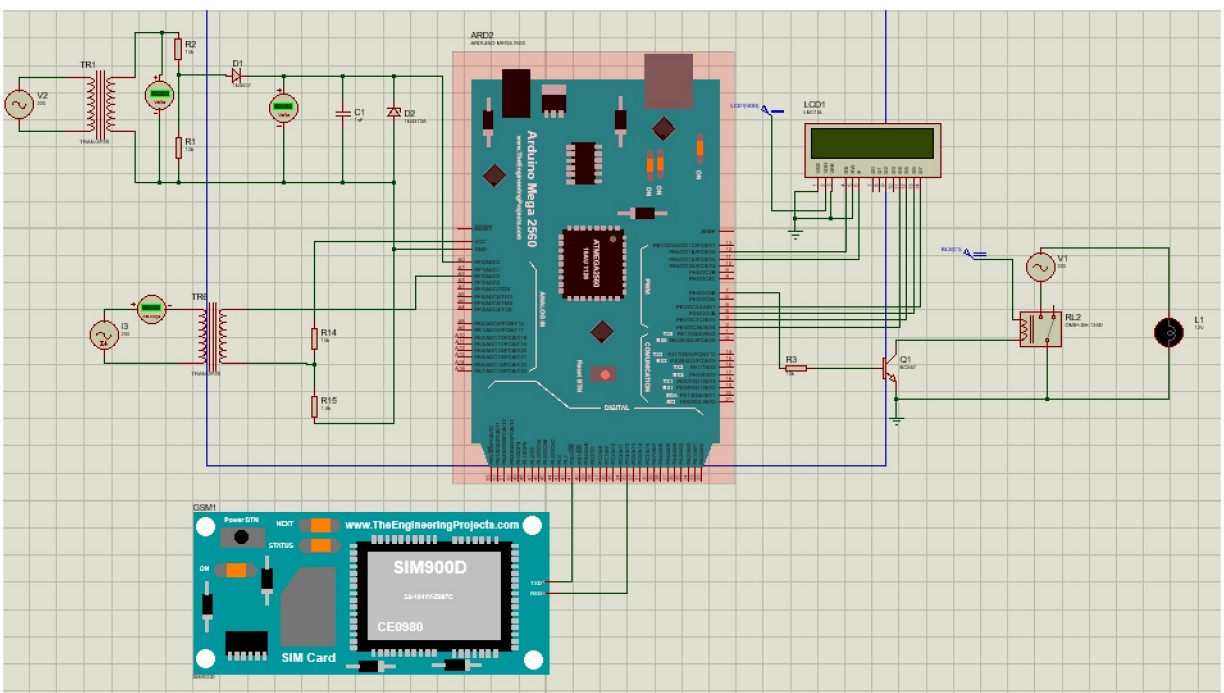

**Fig 8. Proteus simulation.**

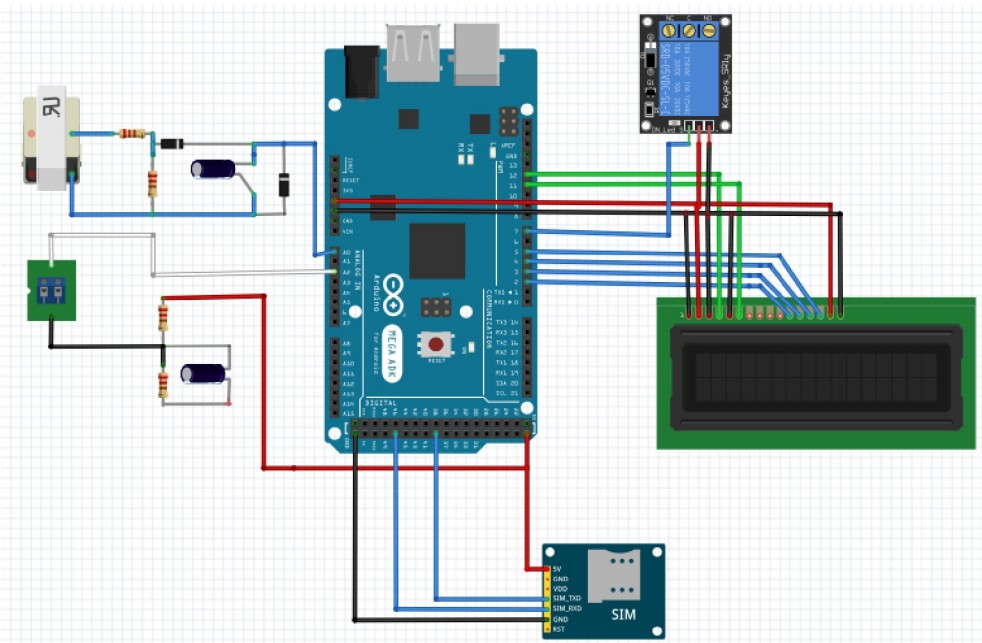

**Fig 9. Fritzing simulation.**

**Energy meter.**   In this research, an electro-mechanical energy meter, as shown in the Fig 12, is selected for measurement and running part of the prototype. In the conventional system of measuring electrical power consumption, one complete disc rotation in an electro-mechanical energy meter is considered one unit or 1 KW-hr of energy consumed.

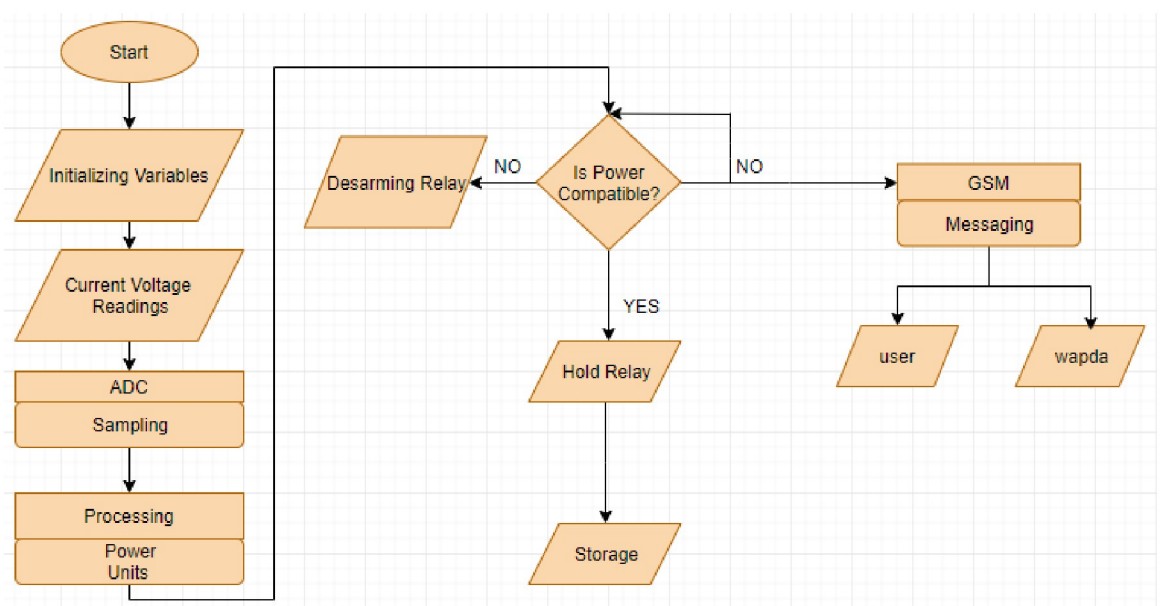

**Fig 10. Flowchart of the research methodology.**

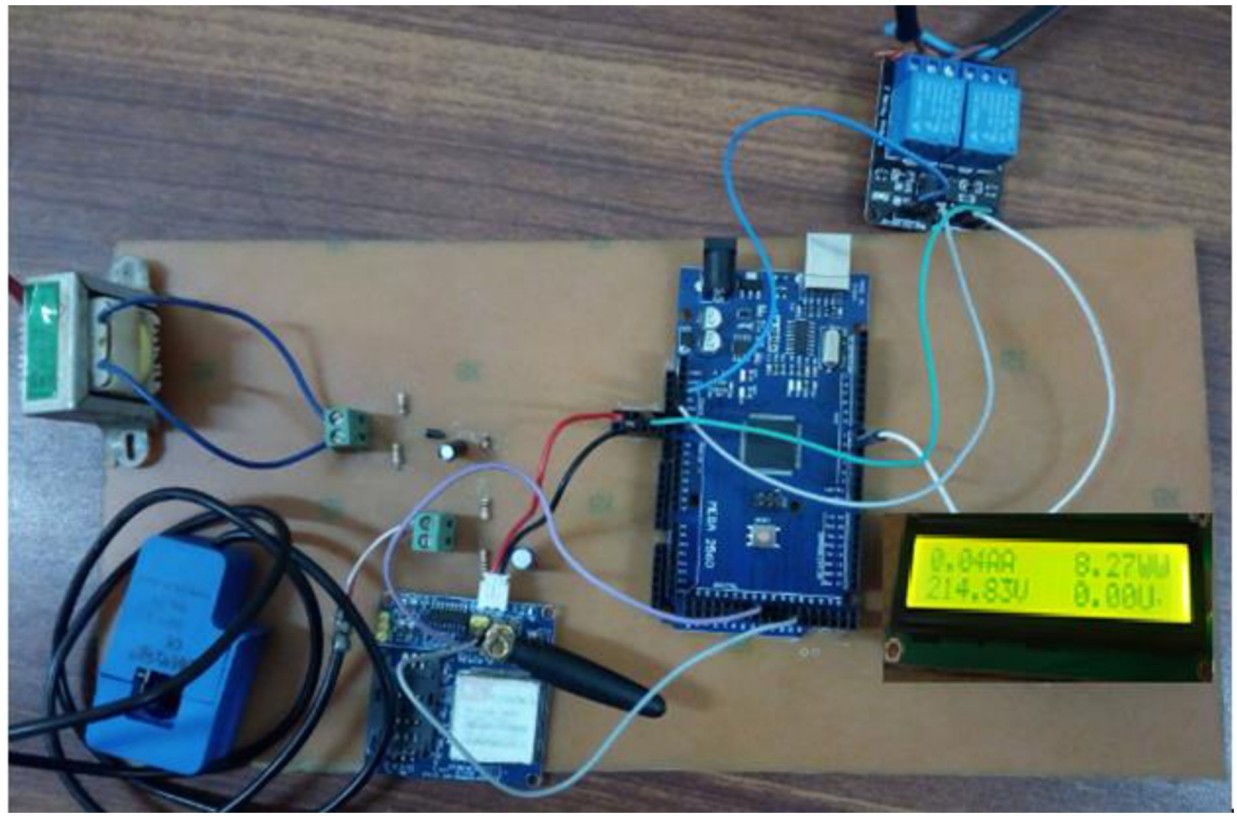

**Fig 11. Hardware prototype.**

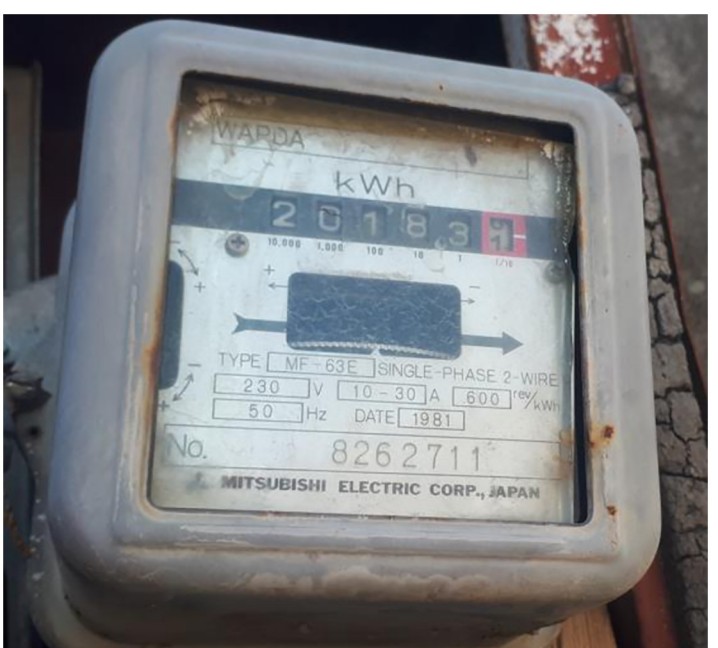

**Fig 12. Energy meter.**

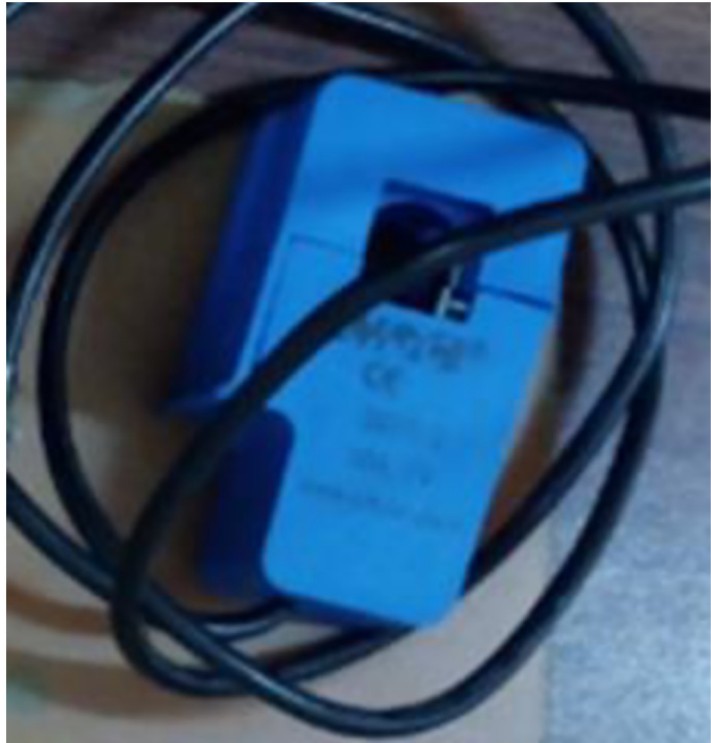

**Fig 13. Current sensor.**

**Current sensor.** The current sensor measures the amount of current flowing in a wire. A wire in which the current is to be measured is passed inside the window of the current sensor. This sensor obeys Hall's effect. For instance, when current flows in a wire, it produces a magnetic field around, which induces another current in the secondary winding of the current sensor. The sensor in the proposed system scales a current of 30 Ampere when 1 Volt is induced, as given in Fig 13.

**Global System for Mobile (GSM) module.** Global System for Mobile communication (GSM) is digital and open cellular technology that transmits data through texts, voices, images, and graphics. It is usually operated at 850MHz, 950MHz, 1800MHz, and 1900MHZ of frequency bands. In this research SIM800A module of GSM architecture is used as shown in Fig 14, Quad-Band with RS232 interface, only responsible component to send a text to both consumer and authority in case of overloading the transformer.

**Relay module.** A relay is a switch that can be operated electrically. It may have one or more terminal inputs and a set of operating contact terminals. The contact terminals are usually named make contacts, break contacts, and their combination. Relay is used for controlling purposes, usually done through signals. In the proposed methodology, the controlling function is performed by the 2-Relay module given in the Fig 15.

**Arduino microcontroller.** Arduino is an open-source low-cost, easy-to-use, and flexible programmable microcontroller board that can be integrated with several electronic projects. It is based on a Microchip ATmega328P, which can be interfaced with Raspberry PI, relays, LEDs, motors, and other Arduino boards and shields. This work uses Arduino ATmega2560, also called Mega, as illustrated in Fig 16. It contains 54 digital and 16 analog input/output pins, a 16MHz crystal oscillator, a power jack, a USB connection, and a reset button. It is compatible with almost all designed shields for Arduino by possessing these many qualities.

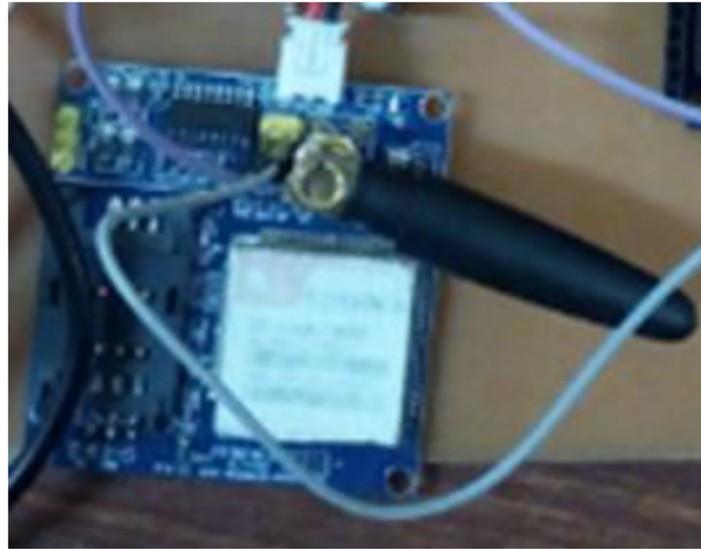

**Fig 14. GSM module (SIM800A).**

In this proposed system, the code is burnt into this microcontroller using Arduino-1.6.12 software.

**Liquid Crystal Display (LCD).** Liquid Crystal Display is an electronic module used for display in a wide range of small devices like wristwatches to big devices like calculators, phones, televisions, computers, etc. It is very beneficial to use as this module is programable and inexpensive, with no limitation on displaying characters, even special animations. In this proposed system, display function is carried out through a 16x2 liquid crystal display, as portrayed in Fig 17.

## Results and discussion

It was examined under different load conditions to check and test the accuracy of the proposed system. Different voltage readings in Table 4, current in Table 5 and power given in Table 6

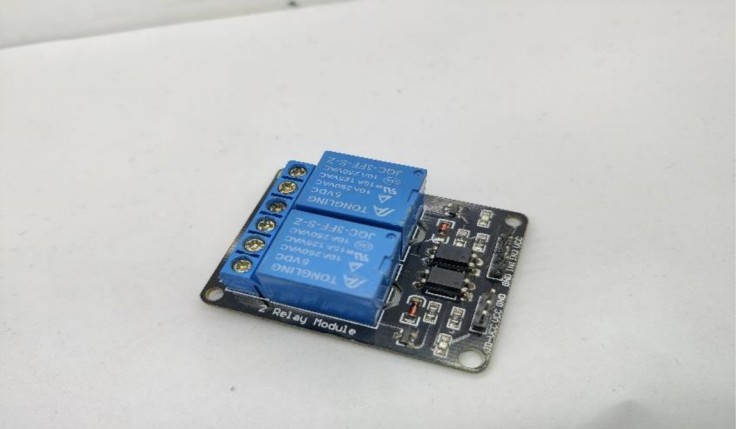

**Fig 15. 2-Relay module.**

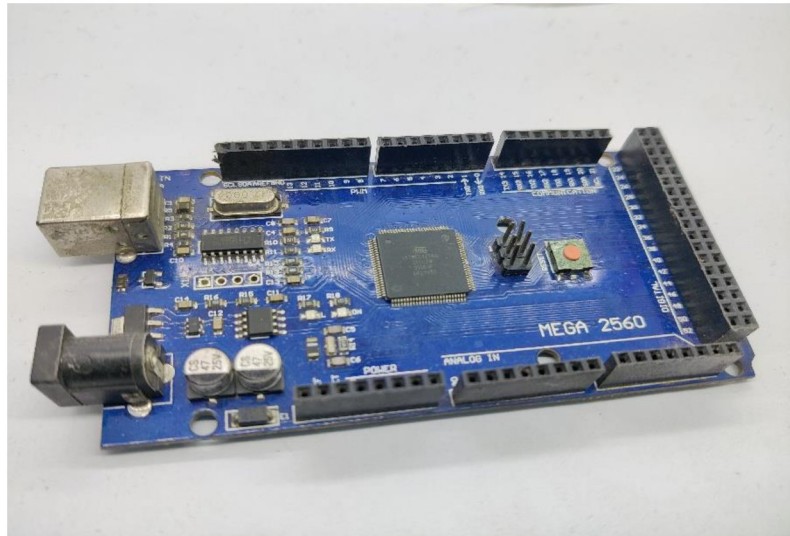

**Fig 16. Arduino ATmega2560.**

are obtained. The results were compared with the measures taken on the traditional energy meter with the same time and applied load as described in real-time running prototype in Fig 18.

From these results, it can be summarized that the proposed methodology can yield a result approximately similar to the conventional energy meter. But there is still some deviation in the results, showing almost an error from 0.6% to 0.9%. Hence the results graphed here assure the proposed methodology is better as almost 99% as the error is less than or equal to 1%. Fig 19 shows the message sent by the authority as a warning to the consumer. Fig 20 shows the information to the authority by the prototype.

## Comparison of the proposed system with the existing smart meters

The developed system is compared with the readily available smart meter in the market in Table 7.

The developed system meets the basic requirements essential for consumers. In addition, the developed system provides remote supply connectivity, disconnectivity, and configuration

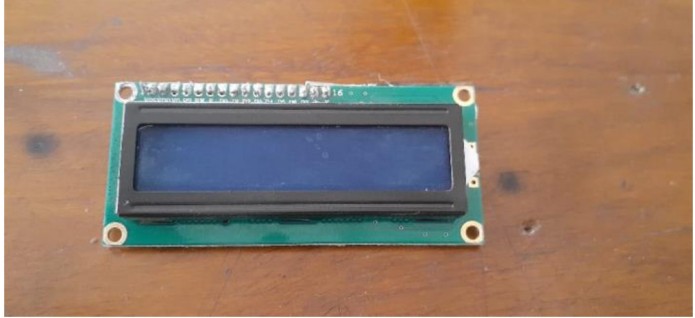

**Fig 17. Liquid Crystal Display (LCD).**

**Table 4. Voltage readings.**

| Readings | Expected Reading | Measured Reading |
|---|---|---|
| 01 | 220 | 219 |
| 02 | 220 | 218 |
| 03 | 220 | 218.5 |

**Table 5. Current readings.**

| Readings | Expected Reading | Measured Reading |
|---|---|---|
| 01 | 0.45 | 0.47 |
| 02 | 0.92 | 0.95 |
| 03 | 2.72 | 2.86 |

**Table 6. Power readings.**

| Readings | Expected Reading | Measured Reading |
|---|---|---|
| 01 | 99 | 102.9 |
| 02 | 198 | 204 |
| 03 | 598 | 612 |

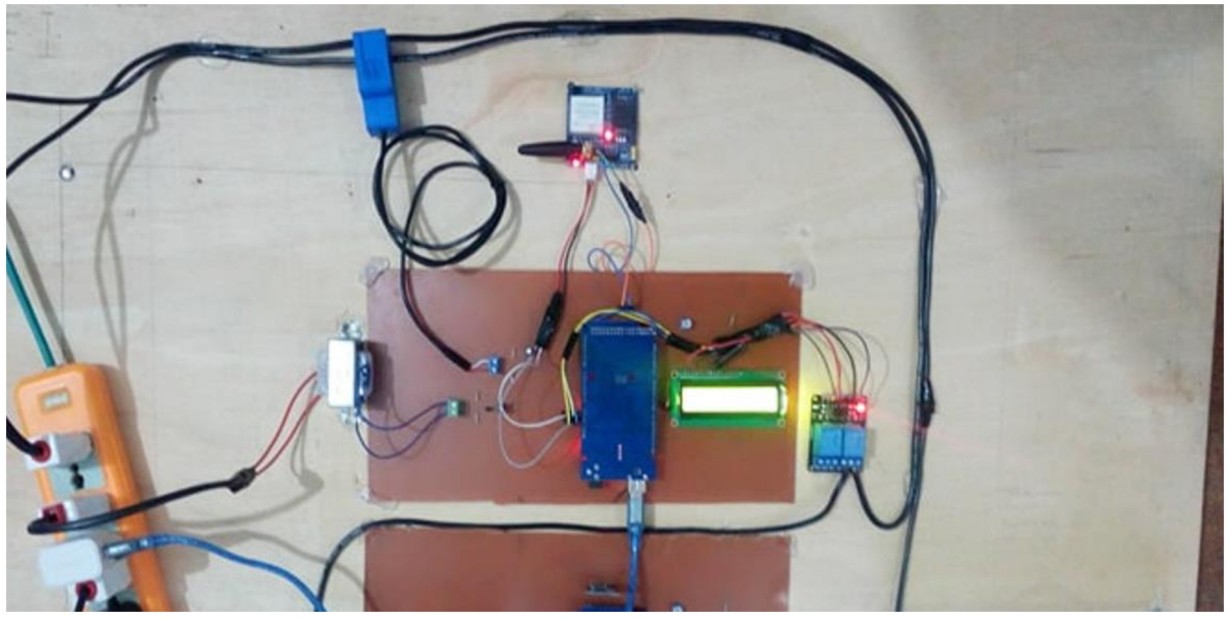

**Fig 18. Real time running prototype.**

capability in contrast to the existing smart meter. Furthermore, the developed system provides a cost-efficient solution to the consumer by 61%. The conventional mindset is one of the major hindrances in adopting the concept. Still, the lesser cost associated with the developed system makes it an ideal solution for the consumer.

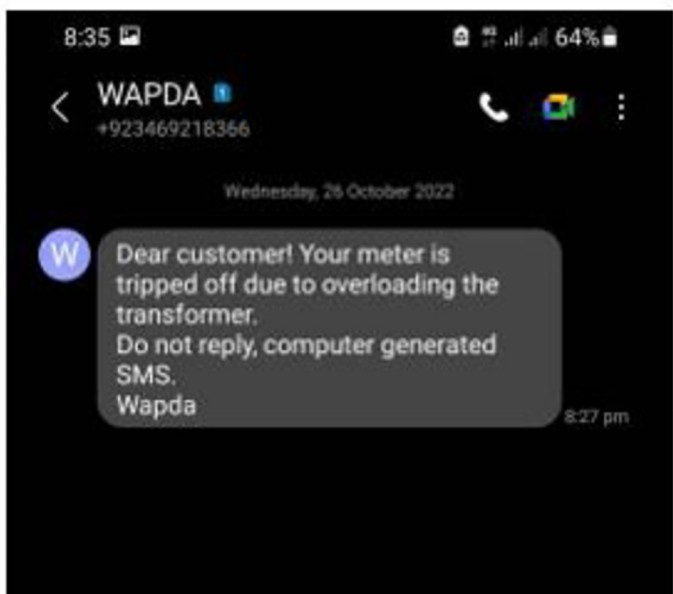

**Fig 19. SMS to the consumer as warning.**

## Conclusion

The central aim is to provide a consumer-friendly electric energy measuring system with the least human fatigue and labour, with the help of implementing modern techniques and technology which is even integrable with the traditional system. Different sensors measure the amount of power consumed by an entity. If the power dragged more than the prescribed amount, the Arduino microcontroller is there to cut off the main line of the consumer by actuating the 2-Relay module. This system also employs the GSM module to keep bidirectional communication strong between customers and authorities. The authority is made powerful as this device helps it get notified in case of unwanted situations. Furthermore, energy management is also being practised as this system can be graded, and the desired amount of power can be allotted to a customer. The proposed system provides an energy-friendly and

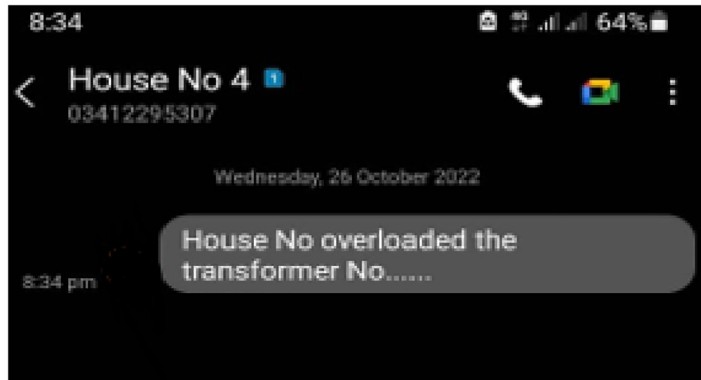

**Fig 20. SMS to authority as information.**

**Table 7. Comparison of the developed system with already available smart energy meter.**

| Points of Consideration | Developed System | Readily Available Smart Meters in the market [37] |
|---|---|---|
| Reading and Billing remotely | ✓ | ✓ |
| Remote Supply Connection and Dis-Connection | ✓ | ✗ |
| Reconfiguration Remotely | ✓ | ✗ |
| Display of Information | ✓ | ✓ |
| Accuracy | 0.9–0.99% | 1% |
| Cost | $20.76 | $122.37 |

sustainable environment within the community and can be helpful to keep Pakistan standing among the row of electrically-smart nations.

## Future work

The proposed system can be customized for further objectives like;

- Online Billing System

- Pre-paid Billing System

- Power Theft Protection

- Commercial Usage

This can be beneficial enough for the industries to eliminate the penalties due to the low power factor.

## Author Contributions

**Conceptualization:** Shahbaz Khan.

**Data curation:** Mian Hazrat Shah, Shahbaz Khan.

**Investigation:** Asif Khan.

**Resources:** Sayed M. Eldin.

**Supervision:** Ilyas Khan.

**Writing – original draft:** Mian Hazrat Shah, Shahbaz Khan.

**Writing – review & editing:** Ilyas Khan, Sayed M. Eldin.

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
