## [Decision Letter · Decision Letter 0]

7 Feb 2023

PONE-D-23-00228Sustainable Energy management Using Internet of Things (IoT)PLOS ONE

Dear Dr. Khan,

Thank you for submitting your manuscript to PLOS ONE. After careful consideration, we feel that it has merit but does not fully meet PLOS ONE’s publication criteria as it currently stands. Therefore, we invite you to submit a revised version of the manuscript that addresses the points raised during the review process.

We look forward to receiving your revised manuscript.

Kind regards,

Amit Kumar

Academic Editor

PLOS ONE

Journal Requirements:

2. Please note that PLOS ONE has specific guidelines on code sharing for submissions in which author-generated code underpins the findings in the manuscript. In these cases, all author-generated code must be made available without restrictions upon publication of the work. 

Please review our guidelines at https://journals.plos.org/plosone/s/materials-and-software-sharing#loc-sharing-code and ensure that your code is shared in a way that follows best practice and facilitates reproducibility and reuse.

5. We note that Figure 4 and 11 to 18 in your submission contain copyrighted images. All PLOS content is published under the Creative Commons Attribution License (CC BY 4.0), which means that the manuscript, images, and Supporting Information files will be freely available online, and any third party is permitted to access, download, copy, distribute, and use these materials in any way, even commercially, with proper attribution. For more information, see our copyright guidelines: http://journals.plos.org/plosone/s/licenses-and-copyright.

(1) You may seek permission from the original copyright holder of Figure 4 and 11 to 18 to publish the content specifically under the CC BY 4.0 license. 

6. Please ensure that you refer to Figure 1, 5, 6, 15 to 18 in your text as, if accepted, production will need this reference to link the reader to the figure.

7. We note you have included a table to which you do not refer in the text of your manuscript. Please ensure that you refer to Table 1, 3 to 5 in your text; if accepted, production will need this reference to link the reader to the Table.

Reviewers' comments:

Reviewer's Responses to Questions

**Comments to the Author**

1. Is the manuscript technically sound, and do the data support the conclusions?

Reviewer #1: No

Reviewer #2: Yes

2. Has the statistical analysis been performed appropriately and rigorously? 

Reviewer #1: Yes

Reviewer #2: Yes

3. Have the authors made all data underlying the findings in their manuscript fully available?

Reviewer #1: Yes

Reviewer #2: Yes

4. Is the manuscript presented in an intelligible fashion and written in standard English?

Reviewer #1: Yes

Reviewer #2: Yes

5. Review Comments to the Author

Reviewer #1: This research focuses on energy management, making the distribution authority more powerful, digitalization and protection of expensive components in electrical power systems. The proposed methodology uses current and voltage sensors to remotely monitor the amount of power being supplied to the consumer continuously, along with a microcontroller responsible for activating the relay in case of over-consumption and the Global System for Mobile (GSM) network to warn the consumer and inform the authority.

In general, there is no scientific contribution in this paper. The authors are encouraged to clarify the constraints and the cost of practical implementation for each consumer.

Reviewer #2: A real time working model as mentioned can be added with some snapshots of the device working with GSM.

Cost specification to be clear on a broader end of usage.

Software utilized specification to be added.

Critical review of previous available works/ relative works can be added to increase the application feature.

6. PLOS authors have the option to publish the peer review history of their article (what does this mean?). If published, this will include your full peer review and any attached files.

Reviewer #1: **Yes: **CHUN-LIEN SU

Reviewer #2: **Yes: **Dr.N.Vijayalakshmi

---

## [Author Response · Author response to Decision Letter 0]

21 Feb 2023

We would like to extend our warm regards to the reviewers for sharing their expert opinions and useful comments. 

This is indeed a great pleasure for us to see that all the reviewers are overall positive about the manuscript. The comments suggested by the reviewers have helped us to further improve the paper. Here are the point-wise details of the revision. The response has been highlighted in blue font.

Reviewer 1:

1. “This research focuses on energy management, making the distribution authority more powerful, digitalization and protection of expensive components in electrical power systems. The proposed methodology uses current and voltage sensors to remotely monitor the amount of power being supplied to the consumer continuously, along with a microcontroller responsible for activating the relay in case of over-consumption and the Global System for Mobile (GSM) network to warn the consumer and inform the authority. In general, there is no scientific contribution in this paper. The authors are encouraged to clarify the constraints and the cost of practical implementation for each consumer”.

The authors would like to thank the reviewer for the positive remarks and constructive feedback. As suggested by the reviewer, the authors have revised their manuscript significantly. As suggested, the revised manuscript's main contribution has been highlighted through track changes. The revised text has also been reproduced below verbatim.

“The main contribution of this proposed research is to protect the expensive elements like a transformer from consumer’s side, while considering sustainability and not using chemicals, gases, or oil on transformer’s side.”

The authors have also compared the developed system with the readily available smart meters in the market, as shown in table 7 of the revised manuscript. The main constraint in adopting the concept is the conventional mindset of the consumer. However, the developed system is significantly cheaper than the existing smart meters.

Reviewer 2:

“A real time working model as mentioned can be added with some snapshots of the device working with GSM. Cost specification to be clear on a broader end of usage. Software utilized specification to be added. Critical review of previous available works/ relative works can be added to increase the application feature”.

The authors would like to thank the reviewer for their appreciation. The reviewer's comments have helped us further improve the paper. The authors have compared the developed system with the existing smart meters in table 7 of the revised manuscript. The cost specification of the developed system and its comparison with the readily available smart meter is also highlighted in the revised manuscript through track changes.

The specification of the software utilized has also been added in the revised manuscript and highlighted through track changes.

The critical review of the literature has been summarized in table 3 of the revised manuscript and highlighted through track changes.

---

## [Decision Letter · Decision Letter 1]

16 Mar 2023

Sustainable Energy management Using the Internet of Things (IoT)

PONE-D-23-00228R1

Dear Dr. Khan,

We’re pleased to inform you that your manuscript has been judged scientifically suitable for publication and will be formally accepted for publication once it meets all outstanding technical requirements. Sorry for the delay. I invited the previous two reviewers, and only one of them accepted to the invitation to review the revised paper, which caused the delay.

Kind regards,

Amit Kumar

Academic Editor

PLOS ONE

Additional Editor Comments (optional):

Reviewers' comments:

Reviewer's Responses to Questions

**Comments to the Author**

1. If the authors have adequately addressed your comments raised in a previous round of review and you feel that this manuscript is now acceptable for publication, you may indicate that here to bypass the “Comments to the Author” section, enter your conflict of interest statement in the “Confidential to Editor” section, and submit your "Accept" recommendation.

Reviewer #2: All comments have been addressed

2. Is the manuscript technically sound, and do the data support the conclusions?

Reviewer #2: Yes

3. Has the statistical analysis been performed appropriately and rigorously? 

Reviewer #2: Yes

4. Have the authors made all data underlying the findings in their manuscript fully available?

Reviewer #2: Yes

5. Is the manuscript presented in an intelligible fashion and written in standard English?

Reviewer #2: Yes

6. Review Comments to the Author

Reviewer #2: A revision on critical analysis has been done and could be appreciated. Software analysis was also added and which could be accepted.

7. PLOS authors have the option to publish the peer review history of their article (what does this mean?). If published, this will include your full peer review and any attached files.

Reviewer #2: No

---

## [Editor Report · Acceptance letter]

28 Mar 2023

PONE-D-23-00228R1 

Sustainable Energy Management Using the Internet of Things (IoT) 

Dear Dr. Khan:

I'm pleased to inform you that your manuscript has been deemed suitable for publication in PLOS ONE. Congratulations! Your manuscript is now with our production department. 

Kind regards, 

on behalf of

Dr. Amit Kumar 

Academic Editor

PLOS ONE